# Effect of organizational change on employee innovation performance: A dual mediation model

Teng Liu[1], Hao Wang[1], Yaru Liu[2], Zhenzhu Li [1]*, Yiting Zhang[2], Honghong Zhu[2], Lei Ning[2], Daokui Jiang[2]

**1** Shandong Labor Vocational and Technical College, Jinan, China, **2** Business School, Shandong Normal University, Jinan, China

\* 297604920@qq.com

## Abstract

Affected by the COVID-19 pandemic and the international development pattern, the international environment has undergone profound changes. Enterprises, as the main body of activities on the front line of production and operation and the main battlefield of market competition, are facing various risk challenges. In both domestic and international markets, these challenges are becoming increasingly complex for businesses to navigate. For theoretical research, the impact of organizational change on employee innovation performance has become a key issue in organizational behavior and human resource management research. However, the influence mechanism of organizational change on employee innovation performance is still unclear. In this study, we examine whether, how, and when organizational change increases employee innovation performance in accordance with job demands-resource theory, as well as the effect of work pressure and work engagement on employee innovation performance. Data from 289 employees at three time points are examined. The results show that: (1) Organizational change negatively affects employee innovation performance through work pressure, i.e., work pressure mediates the impact of organizational change on employee innovation performance. (2) Organizational change positively affects employee innovation performance through work engagement, i.e., work engagement mediates the impact of organizational change on employee innovation performance. (3) Organizational identity plays a moderating role between organizational change and work pressure and work engagement, respectively, and there is a moderating effect in the process of mediation of work pressure and work engagement. The findings of this study provide important insights into how and when organizational change influences employee innovation performance.

## Introduction

In recent years, the focus on innovation performance has moved to the center stage of scholarly research on Human Resource Management (HRM) [1]. Innovation performance is an

**Competing interests:** The authors have declared that no competing interests exist.

important indicator that must be paid attention to in building the long-term sustainability of enterprises, especially in the context of VUCA (Volatile, Uncertain, Complex, Ambiguous) era, and the current external competition faced by enterprises is increasingly intense. The rapid iterations of the digital economy and unpredictable technological changes have increased the competitive pressure on enterprises, and how to achieve the transformation and upgrading of enterprises and digital empowerment with the help of innovation has become a key topic of concern in the industry. As the most active economic unit, enterprises, by pooling superior resources, creating an innovation system and improving innovation efficiency, are important measures to build up their competitive advantages and achieve long-term sustainable development. Based on this, it is of great practical significance to explore the influencing factors of enterprise innovation from a microscopic perspective. At present, many scholars have studied the antecedents influencing the innovation performance of firms from various perspectives such as open search strategies, dedicated human capital [2], and innovation networks [3]. So, are there other factors that may have an impact on employee innovation performance?

Most studies have concluded that organizational change is a stressor for employees at work and the effect of change on employees depends on their stress coping strategies [4]. In fact, organizational change may also motivate employees to work and promote positive and positive behaviors. The rapidly changing environment leads to more frequent and urgent organizational changes, even in time to prepare for them, and employees may first face changes in their job characteristics when meeting organizational changes, which often directly affect employees' work attitudes and thus have a differential effect on innovative performance. The JD-R model states that when the job generates more work demands, employees tend to feel tired due to higher stress and have a negative effect on work When work resources are increased, employees are not only relieved of work pressure, but also motivated, motivated, and more engaged in their work, which in turn produces positive work outcomes [5]. Therefore, changes in work characteristics brought about by organizational change may have negative and positive dual effects on employees' innovation performance through both work pressure and work engagement paths, respectively, and the investigation of this mechanism is an important guide for enterprises to improve employees' innovation performance in the context of organizational change [6].

The success of organizational change is closely related to employees' attitudes and behaviors. In other words, whether employees actively participate and respond to organizational change largely determines whether the change can be successfully promoted and achieve good results, while the relationship between employees and the organization can largely influence employees' attitudes and performance in the change. Organizational identity is a kind of cognitive and emotional belonging that employees regard themselves and the organization as a unified body [7]. When employees identify themselves as members of an organization, they will make favorable tendencies toward the organization in terms of resources, behaviors, and emotions, etc. Employees with high organizational identity tend to show a supportive attitude toward the organization or are more willing to maintain consistency with organizational goals in their actions. In particular, in the context of organizational change, employees with high levels of organizational identity tend to show positive attitudes toward change and are able to respond positively and actively participate in their actions, stimulating their personal potential and creativity to facilitate the smooth implementation of change and the achievement of organizational goals. Therefore, the differentiated level of organizational identity may play a boundary-binding role in the process of the dual-path influence of organizational change on employees' innovation performance.

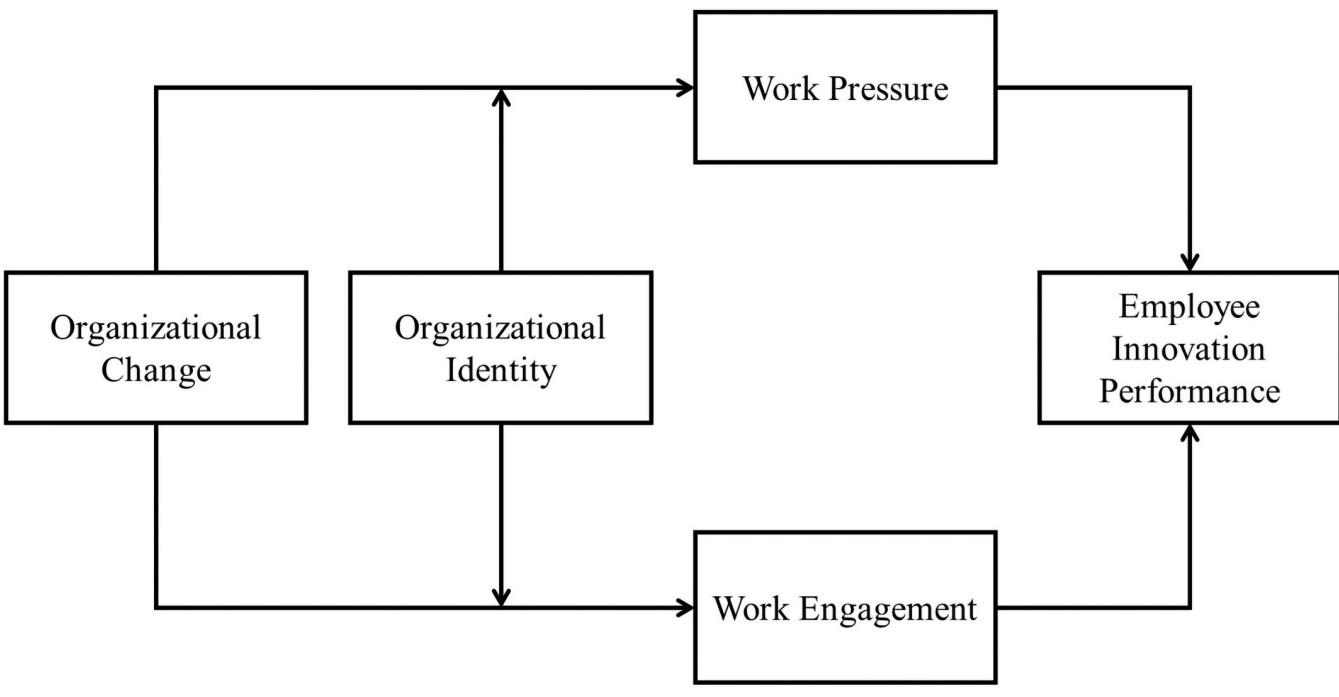

**Fig 1. Research model.**

By testing the proposed research model (see Fig 1) the study contributes to the HR, organizational change, and innovation performance literature in two primary ways. First, explores the mechanism of the influence of organizational change on employees' innovation performance, which enriches the research on the influence of organizational change on employees' work outcomes and expands the antecedent research on employees' innovation performance. The JD-R model has been widely cited and studied by more scholars since its introduction, but the existing literature rarely applies this model theory to organizational change contexts, ignoring the relationship between organizational change and employee performance based on the JD-R model perspective. While past studies have extensively explored employee innovation performance, focusing more on the individual employee and work level, the number of studies on the relationship between organizational change, as a special organizational contextual factor, and employee innovation performance is somewhat stretched. In doing so, it directly responds to calls from researchers to investigate possible organizational factors, i.e., organizational change, which can affect employees' innovative performance. The second contribution of the study is that it investigates the boundary conditions of the organizational change-innovation performance link. This is critical to decipher whether organizational change can universally increase or reduce innovation performance or whether their impact is dependent on contextual factors. A number of authors have called for research which explores the important variables capable of enhancing or undermining the effectiveness of organizational change [8] and in this regard we look at a critical albeit unexplored factor which occurs inside the domain of work i.e. organizational identity.

In the following sections, we discuss the links in the proposed model and the theoretical underpinnings for the proposed hypotheses. Then the paper will discuss the employed methodology before discussing the results and implications for both theory and practice.

## Theoretical background and hypotheses

### Job demands-resources theory

The Job Demand-Resource (JD-R model) is a theoretical achievement formed by Demerouti et al by drawing on the ideas of resource conservation theory, job demand-control theory [5], and give-and-take imbalance. Once the model was proposed, it received wide attention from the fields of management and psychology, and scholars have conducted rich studies based on the JD-R model, which have enriched the content of the model and helped in its continuous improvement and revision [9]. At present, the JD-R model has developed into a more mature and perfect JD-R theory [10].

According to the JD-R model, any occupation possesses unique factors that affect employee health and work status, all of which can be attributed to job demands and job resources, which together constitute job characteristics [11]. Among them, work demands are those factors in work that involve physical, psychological, social or organizational aspects that require continuous physical and mental effort, which are related to physical and mental exertion, such as work-family conflict, role stress, work load, etc. Work resources are the resource factors that can provide workers with physical, mental, social or organizational support and assistance, such as autonomy, supervisor support, work participation, performance feedback, etc. These resources help motivate workers to achieve their goals, reduce the physical and psychological exertion generated by work demands, and promote personal growth, learning and development.

The JD-R model proposes a dual process of positive and negative effects of job characteristics on employees' job outcomes, which provides the theoretical basis for this paper. In the process of organizational change, employees usually face role change, job content or way adjustment, etc., which leads to changes in job requirements and job resources. Continuous or high level of change requirements can damage employees' physical and mental health and lead to resource depletion, causing psychological stress and job burnout, which negatively affect their innovation performance. However, the abundant supply of change resources can alleviate the negative effect of change requirements, motivate employees to work, motivate employees to actively participate in their work, and thus improve their innovation performance. Therefore, in the process of organizational change, employees may perceive changes in job requirements and job resources and produce negative and positive dual psychological and behavioral outcomes, respectively, which have differential effects on their individual innovation performance.

**Social identity theory.** Social identity theory, proposed by Tajfel and Turner among others, is a highly influential foundational theory in the study of intergroup relations [12]. The theory posits that social identity is the individual's cognition of belonging to a particular social group, as well as the emotional and value significance obtained as a member of that group [13]. An individual's social identity is primarily established and developed through social categorization, social comparison, and the principle of positive distinctiveness [12]. Social categorization refers to the spontaneous classification of people and things into in-groups and out-groups, attributing in-group characteristics to oneself, and concentrating more favorable resources within the in-group. Social comparison refers to the tendency to exaggerate inter-group differences to some extent and to evaluate in-group members more positively when comparing between groups. The principle of positive distinctiveness suggests that individuals, when part of a group, are motivated to differentiate themselves by performing better than other group members.

Building on social identity theory, the concept of organizational identity has gradually emerged and developed, providing a theoretical basis for explaining employee

organizational socialization. In 1989, Ashforth and Mael [14] expanded social identity theory in the field of organizational research and proposed a definition of organizational identity, which is the individual's consistency with the organization or the perception of belonging to an organization. Employees with high organizational identity can share the organization's honor and disgrace, fully support and maintain the organization, and actively strive for the organization's benefits [15].

Organizational identity is the employee's recognition and belonging to the organization, which plays a positive role in improving employee job satisfaction, promoting organizational citizenship behavior, and reducing employee turnover intentions [16]. Organizational identity theory elaborates on the formation process of organizational identity and the possible paths for employees to build organizational identity from different perspectives. This provides a theoretical basis for explaining the boundary role of organizational identity in employees' work psychology and behavioral changes, and is instructive for helping to build or improve the level of employee organizational identity to enhance employee and organizational performance.

## Organizational change, work pressure and employee innovation performance

**Organizational change and work pressure.** Organizational change is a process of strategy, structure, technology, human resources and other factors change carried out by an organization to better achieve its goals. Although organizational change takes the improvement of business operation as the starting point, there is significant evidence that organizational change will lead to work pressure among employees [17]. JD-R model points out that the work requirements of employees at work will lead to the loss of individual health, continuous or high level of work requirements may consume employee resources, and then consume mental energy and cause stress process. As a result, employees may feel higher levels of work pressure in the face of increased job demands.

Research shows that organizational change will lead to increased job requirements of different degrees. The process of organizational change is often accompanied by organizational structure adjustment, system adjustment, technology upgrade, personnel adjustment and other actions. In order to ensure the smooth implementation of the change, the organization usually promulgates new organizational policies and systems and establishes new rigorous working procedures. In this process, Organizational politics and red tape in organizations may consume more energy and resources of employees and increase their sense of stress [18]. Furthermore, in order to adapt to the changes generated by the reform, employees need to understand new organizational policies, learn new routine procedures and skills. These work requirements will increase the workload and cognitive impairment of employees, leading to work pressure. In addition, employees may encounter the change of job positions and the change of superior leaders in the process of reform, and need to establish working relationships with new colleagues, which undoubtedly increases their requirements in interpersonal communication, and even leads to role conflicts and ambiguities, and interpersonal conflicts. In addition, many studies have pointed out that uncertainty is also the main source of pressure of organizational change on employees, and employees will try to search for information about change-related events to understand the content of the change to reduce uncertainty, thus generating work pressure [19]. Organizational change is often accompanied by the implementation of new technologies, organizational structure adjustment, relocation, mergers and acquisitions, and personnel reform, among which the frequency and intensity of change may also increase the working pressure of employees and undermine their well-being [20].

Based on the above analysis, this paper proposes the hypothesis that

H1a: Organizational change positively affects work pressure.

**Work pressure and employee innovation performance.** Once work pressure is generated, employees' work emotions, attitudes and behaviors are bound to be affected. Most scholars believe that the effects of work pressure on employees include both positive and negative aspects, which are related to the dual attributes of challenging and hindering stress. Research on the effects of the double-edged sword of work pressure from the perspective of the dual attributes mainly reflects the positive results of challenging stress and the negative results of hindering stress. Organizational change usually implies workload, job uncertainty, organizational politics, and cumbersome work procedures, among which hindering stress is more significant than challenging stress, so scholars prefer hindering stress in the classification of stressors [21,22]. Obstructive stress refers to stresses that employees find difficult or impossible to cope with, which are difficult for employees to resolve through their own efforts and therefore prevent them from self-improvement and goal achievement. Although the challenging pressures of organizational change can motivate employees to work, challenge themselves, and increase their work engagement, the accompanying hindering pressures have a more significant impact on employees. Obstructive pressures do not have potential benefits and not only hinder employees' work process, but also consume their resources, so employees do not invest more time and energy in searching for new ways to solve problems, but rather "settle" for the existing solutions. Once motivation for innovation is weakened, innovation performance is naturally lower [23]. A number of empirical studies have confirmed these inferences. Researchers based on a cognitive resource theory perspective, argued that work pressure reduces the cognitive resources needed to generate innovative thinking and leads to a greater tendency to habitual patterns of thinking and action, which undermines creativity. Other studies have confirmed the idea that work pressure inhibits individual innovation [24].

Based on the above analysis, this paper proposes the hypothesis that

H1b: Work pressure negatively affects employees' innovation performance.

**The mediating role of work pressure.** Organizational change is implemented and advanced with a new set of events, and employees involved in the change or new event are usually asked to think and act in new ways to engage in their work, and in this context they may feel threatened and more stressed [25]. Organizational change requires employees to learn new skills, work processes, and adapt to a new culture, which can increase the work demands of employees, and demanding work consumes employees' mental and physical resources and leads to negative outcomes by draining energy [5,26]. As employees perceive organizational change as a stressor of increased work demands, they tend to evaluate organizational change negatively, generate negative emotions, behaviors, and attitudes toward organizational change, reduce commitment and trust in the organization, and increase cynicism [27]. When employees have difficulty in anticipating the benefits of organizational change, they reduce the amount of time and energy they devote to their work to protect their mental and physical resources, and exhibit a "status quo" attitude toward work, which does not facilitate the generation of novel ideas in their work. When employees' motivation and initiative to explore new things is weakened, their level of innovation performance is reduced.

Based on the above analysis, this paper proposes the hypothesis that

H1c: Organizational change has a negative impact on employee innovation performance through work pressure.

## Organizational change, work engagement and employee innovation performance

**Organizational change and work engagement.** The JD-R model states that work resources motivate employees, enhance work engagement, and help employees achieve superior performance [5]. Change researchers have confirmed that information about change, participation in change, supervisor support for change, decision-making autonomy, and career development opportunities are positively associated with positive employee feedback on organizational change, and that an increase in work resources related to organizational change facilitates increased employee change engagement [28]. Organizational change is a series of activities in which orders are given from the top down and feedback is given from the bottom up, and in order to promote successful implementation of change, leaders at all levels often provide supportive resources to employees, including a clear purpose and vision for change, clear communication about the change message, and positive encouragement and assistance with the change [29]. Employees who feel supported by their superiors are more motivated to commit to their work and show full energy and stronger dedication. In addition, change involvement is an important work resource in organizational change. It has been shown that employees who have the opportunity to participate in the design and implementation of change are more willing to actively support the change and are more engaged in the change process [30]. Researchers showed, based on the JD-R model, that a work environment that provides employees with job control during organizational change also enhances employees' willingness to dedicate their personal capabilities to their work, increases employee work commitment, and induce employees to evaluate organizational change favorably [31].

In addition to this, other scholars have focused on procedural justice as a working resource in organizational change. Organizational change is a complex process in which organizations select change agents to gather relevant information about employees' concerns and create new rules and mechanisms (i.e., procedural justice) to support the change. A high level of procedural justice means that employees have more opportunities to express their opinions and can use such work resources to protect themselves from the effects of change. Particularly in the context of organizational change, action initiatives in which companies seek to listen to employees and provide relevant information can signal organizational procedural fairness to employees, who perceive an increase in work resources and thus have a positive attitude toward organizational change. When employees have more resources, they are also more willing to adjust their behavior to accommodate change and enhance their work engagement during change [32].

Based on the above analysis, this paper proposes the hypothesis that

H2a: Organizational change positively affects work engagement.

**Work engagement and employee innovation performance.** The relationship between job attitudes and performance has been the focus of academic inquiry in existing research and practice, where work engagement is considered a stronger predictor of performance than other attitudinal constructs such as job satisfaction, job commitment, and organizational commitment [33]. Work engagement is the physical, emotional, and cognitive involvement of employees into their organizational roles, expressed as energy, dedication, and focus. Cognitive engagement has been shown to be the mental energy required for innovative behavior, and in the process of generating ideas, employees invest additional cognitive effort to support better and different systems and processes through cognitive flexibility [34]. In this process, employees expand their cognitive and perceptual scope by revisiting their existing knowledge

structures and may experiment with nontraditional adaptations and combinations of ideas to mobilize innovative behaviors. Research showed that according to the cognitive-emotional personality system theory, employees are able to complete their work with a sense of responsibility and pride when they are highly engaged in their work, are courageous and persistent and this state of high cognitive activity and arousal can promote their innovative behavior [35].

In addition to cognitive engagement, work engagement includes energetic and dedication, and new employees with higher levels of work engagement are able to enter the workplace more efficiently and show higher levels of engagement and enthusiasm in their work, which in turn leads to more innovative behaviors [36]. From an emotional perspective, employees with high levels of work engagement not only establish a strong emotional connection with their work, but also possess greater confidence to perform the job requirements [37], and even when they encounter various difficulties at work, they are able to proactively find new ways to change the status quo. In this process, novel and practical ideas may thus be elaborated, and employees show greater creativity and higher levels of innovative performance at work. From a physiological involvement perspective, the creative process involvement theory states that the innovation realization process often includes three stages: problem identification, information search, and idea generation, and when employees invest a lot of time and energy in their work, they are highly likely to identify problems and gather information more effectively in their work practices in order to come up with more reliable solutions, thus promoting personal innovation performance.

Based on the above analysis, this paper proposes the hypothesis that

H2b: Work engagement positively affects employee innovation performance

**The mediating role of work engagement.** The process of organizational change is accompanied by the realignment and reallocation of employee work resources, and supervisor support, change participation, and change autonomy can motivate employees to respond positively to change. It has been confirmed that superior support, support from colleagues, family and friends, and job rewards in work resources significantly and positively predict work engagement [5]. In addition, organizational change usually involves changes in technology, management models, or business process re-engineering, and training and development related to change is a practical activity that organizations must consider and implement. By participating in training, employees have the opportunity to gain access to enhance their personal skills development, which has a positive effect on promoting their work engagement during change. In addition, new policies and systems are often enacted during organizational change, and employees' performance under the new rules is of high concern to organizational leaders, who often support and encourage their work engagement and active response during change periods and give timely performance feedback on their performance. Developmental feedback from superiors, as an important work resource for employees, can stimulate motivation and engagement motivation and lead to more passionate commitment to work [38].

When employees are at a high level of work engagement, they are willing to work hard for their jobs and do not easily get tired, and they are able to persevere even in the face of work difficulties, showing a positive emotional state at work [39]. In such a context, when employees are engaged in their work and experience positive emotions, their intellectual and psychological resources are expanded, thus promoting them to actively explore, discover and develop original ideas and try to put innovative ideas into practice [40]. Compared to repressed emotions, active and positive emotional experiences facilitate employees to generate greater creative fluency, increase the level of divergent thinking, and therefore generate higher creativity and achieve higher levels of innovative performance [41].

Based on the above analysis, this paper proposes the hypothesis that

H2c: Organizational change positively affects employee innovation performance through work engagement

## The moderating role of organizational identity

Organizational identity is the cognitive and emotional belonging of employees to the organization based on their individual organizational membership identification, and this identity can motivate employees to think about and guide their personal work behavior from the perspective of organizational interests. As mentioned above, organizational change creates a work environment full of uncertainty and insecurity, and employees may feel more work demands and work pressure in the turbulent environment. At the same time, organizational change is accompanied by an increase in work resources, which promotes employees' work engagement.

Organizational identity, as a special self-construal of the employee-organization relationship, is more than a simple exchange between employees and the organization; it can have a direct impact on employees' work attitudes and behavioral choices. Employees with a high level of organizational identity tend to recognize and support organizational decisions, are willing to expand their perceptions of their work roles, and make personal decisions based on the principle of alignment with organizational goals even in the face of uncertainty [42]; this trust and support for the organization can weaken employees' work pressure due to organizational change and strengthen the positive impact of organizational change on work engagement. Organizational identity is spontaneous, and employees' strong spontaneous identification with the organization helps employees to actively cope with stressors at work, and even if some work is unpleasant and depletes resources, employees can still persist and endure [43], thus weakening the positive predictive effect of organizational change on work pressure. On the other hand, employees' sense of belonging to the organization, pride, and other emotional attachments can lead to positive attitudes toward change, especially when employees identify themselves as part of the organization, they will tend to take actions that are beneficial to the organization's development, actively engage in their work and expect to dedicate themselves to achieving organizational goals, thus enhancing the positive effect of organizational change on work engagement. On the contrary, if organizational identity is at a low level and employees' relationship with the organization lacks cognitive and emotional foundation, a series of change initiatives brought by organizational change may intensify employees' dissatisfaction with the organization, and employees need to consume more resources to cope with the change requirements and generate more work pressure. At the same time, according to resource conservation theory, employees may pay less to their work and reduce their personal work commitment in order to protect their resources.

Based on the above analysis, this paper proposes the hypothesis that:

H3a: Organizational identity negatively moderates the effect of organizational change on work pressure.

H3b: Organizational identity positively moderates the effect of organizational change on work engagement.

Organizational identity, as a strong sense of identification and belonging of organizational members to the organization, provides an emotional basis for employees' innovative behavior, and therefore organizational identity can play an influential role in the relationship between organizational change and innovative performance. In a study by Griffin et al [44], it was found that organizational identity is a necessary condition for employees to achieve high levels

of performance in work situations with high uncertainty. Based on social identity theory, high organizational identity will bring individuals closer to the organization and make employees' personal goals converge with the organizational goals of the company, and even in the face of organizational change, employees can match their perceptions of the organization with their own best under this psychological perception to promote the survival and development of the company through pioneering and innovative work attitudes and continuous innovative work behaviors. A study pointed out that employees who have a strong sense of identification with the organization will stand firmly with the organization even when they are in adversity or encounter more difficulties at work, out of trust and belonging to the organization [45]. This resilience can help employees cope positively with work pressure, alleviate burnout under high pressure, and weaken the negative impact of organizational change on innovation performance through work pressure.

Organizational identity promotes the formation of an implicit psychological contract between employees and the organization, and employees with strong organizational identity will develop real dependence and belonging, and thus spontaneously make efforts to maintain the dignity of the organization, which provides strong motivation and persistence for individual innovative behavior [30]. In the face of a series of changes in organizational change, employees with high organizational identity can show stronger cognitive understanding and inclusiveness, and are willing to put in physical and mental efforts to help the organization achieve its goals, which is reflected in their work as full of energy, energetic, and proactive in stimulating personal innovation potential and creativity to improve work efficiency to better adapt to the change. It has been confirmed that employees tend to show greater creativity when they are in a better organizational climate and have a higher level of organizational identification [1]. Conversely, employees who lack the experience of organizational identity find it difficult to align their personal goals with organizational goals in organizational change, and when faced with stress, they often fail to feel the significance of overcoming challenges and tend to develop pessimistic and negative emotions, which in turn amplify the negative transmission effects of work pressure and hinder the positive feedback effect of work engagement, preventing them from achieving the desired innovative performance.

Based on the above analysis, this paper proposes the hypothesis that

H4a: Organizational identity moderates the mediating role of work pressure between organizational change and employee innovation performance.

H4b: Organizational identity moderates the mediating role of work engagement between organizational change and employee innovation performance.

The research model of this paper is shown in **Fig 1**.

## Method

### Sample

This study was approved by Shandong Normal University's Ethical Review Committee (No. BSSDNU700506, Study Title: Organizational Change on Employee Innovation Performance, from May 1, 2023 to May 30, 2024). Due to the large number of voluntary participants, obtaining written consent was impractical. In this project, the willingness to participate in the research survey was taken as an indication of the subjects' consent. Additionally, keeping written records could potentially affect the anonymity and confidentiality of the research results. The study applied a survey method based on a questionnaire to collect data. Data were collected from companies located in the eastern region of China that are in the midst of change.

Access to the participants was gained through professional and personal contacts of the author (s). This study examined five companies that are undergoing organizational change for targeted questionnaire distribution, focusing on data collection and analysis from employees currently working in enterprises undergoing change (During the pre-survey phase, we determined whether companies were undergoing change, including personnel adjustments (such as layoffs, departmental transfers, and large-scale hiring), business adjustments (changes in business scope, consolidation or discontinuation of certain business activities, and the addition of new business activities), organizational structure adjustments (reorganization of departments, creation of new positions, and elimination of some positions), system changes (significant alterations in management systems affecting current work), changes in management approaches (updates to office systems, digital transformation, and the introduction of information management platforms), corporate changes (changes in ownership, acquisitions, mergers with other companies, and changes in shareholders), and any other unlisted transformations. These changes served as the criteria for distributing questionnaires). The surveyed companies come from different industries within Shandong Province, including healthcare, internet, real estate, tourism, high-tech, and traditional manufacturing. The questionnaire distribution and collection were mainly conducted offline through paper questionnaires, supplemented by online electronic questionnaires. The entire process was divided into three stages, with a three-week interval between each stage, to collect longitudinal temporal data from the same subjects. Before distributing the questionnaires, we first contacted the questionnaire coordinators of the investigated companies to state the research purpose and the subjects needed for the study to obtain corporate questionnaire research permission. Secondly, we communicated the precautions for questionnaire distribution and collection, as well as the timing of the three questionnaire stages, and finally proceeded with data collection. Participants in the study were required to fill in an 11-digit mobile phone number as a unique questionnaire code in each stage of the questionnaire. Considering the sensitivity of the mobile phone number and the privacy issues of the subjects, we promised that all questionnaire data would be used solely for academic research, not to be disseminated or used for other purposes. Moreover, while ensuring that the codes were filled in consistently across the three stages, we allowed subjects who still had concerns to fill in an 11-digit virtual mobile phone number as their personal questionnaire code. Participation was voluntary, and respondents were assured of the anonymity of their responses. In addition, we told the participants that all identifying information would be removed to preserve their anonymity.

To reduce the potential common method biases, we conducted surveys in three different phases separated by 3 weeks. If the time lag is too long, it may mask existing relationships; by contrast, if the lag is too short, memory effects may inflate the relation artificially between variables. In the first stage, we asked employees to report organizational change and demographic characteristics, 350 questionnaires were distributed and 328 were returned, for a 93.71% return rate; three weeks later, in the second stage, employees were asked to report work pressure, work engagement, organizational identity, and demographic characteristics, 328 questionnaires were distributed and 309 were returned, for a 94.20% return rate; three weeks later, in the third stage, employees were asked to employees to report innovation performance and demographic characteristics, 309 questionnaires were distributed and 301 were returned, for a 97.41% return rate. To increase the seriousness of completing the answers, a fee of ¥40 was paid to the participant upon completion each time. After all stages of data collection were completed, all online and offline questionnaire data were integrated and aggregated, and the three phases of questionnaire matching were conducted through the code. After eliminating invalid questionnaires with missing or mismatched questionnaire code numbers, questionnaires with questionable answers, and questionnaires where the subjects' work had not been experienced

or affected by organizational change, a total of 289 valid questionnaires were obtained, with an efficiency rate of 96.01%.

## Measures

All scales were measured on a seven-point Likert-scale from 1 (strongly disagree) to 7 (strongly agree). Scores were created by averaging the relevant items. All the scales used are based on existing measures that have been shown to have sound psychometric properties.

Organizational change scale was adopted from Bartunek et al. and includes 15 items [46]. Respondents indicated how often they experience each state using a scale ranging from 1 (*strongly disagree*) to 7 (*strongly agree*). There are 15 questions on the scale, and representative questions include "This change will have an important impact on my future in the company", "This change has changed the way I work", and "I can participate in the change process". The Cronbach's alpha of the scale was 0.958.

Work pressure scale was adopted from Boswell et al. [47]. There are 5 questions on the scale, and representative questions include "I have multiple projects or tasks at the same time and have a large workload or task" and "I have a large amount of work and tasks to complete within a set time frame". The Cronbach's alpha of the scale was 0.918.

Work engagement scale was adopted from Schaufeli et al. [48]. Seven items were selected to measure the variables, including "I feel energetic and dynamic when I work," "When I work, I forget everything around me," and "At work, I always persevere even when things are not going well," etc. The Cronbach's alpha of the scale was 0.972.

Organizational identity three-dimensional scale was adopted from Miller et al. [49]. There are 6 questions on the scale, and representative questions include "I care deeply about the future of my company," "I find that my values are consistent with the organization's values," and "I would like to devote my future to my company. The Cronbach's alpha of the scale was 0.917.

Employee innovation performance scale was adopted from Wang et al. [50]. There are 7 questions on the scale, and representative questions include "I am open to coming up with new ideas in order to improve the current situation," "I am open to turning innovative ideas at work into practical applications," and "I often look for new ways of working, tools, and ways of doing things to improve my work." The Cronbach's alpha of the scale was 0.970.

Several variables were controlled. Following previous studies [51], gender (0 = female, 1 = male), employee age ($1 \leq 25$ years to 4 = 46 years and over), education (1 = high school degree or less to 4 = master's degree or above), job position (0 = general employees, 4 = senior management), tenure ($1 \leq 1$ year to 5 = over 10 years) and company size ($1 \leq 100$, to 4 = over 1000) are included as control variables in the current study.

## Results

### Descriptive statistics

Correlations among all study variables are presented in Table 1. As shown in Table 1, the control variables were correlated with some of the main variables, respectively, indicating that the choice of control variables was meaningful. Organizational change was positively correlated with work pressure (r = 0.327, p<0.001) and work engagement (r = 0.224, p<0.001), respectively, and similarly, employee innovation performance was negatively correlated with work pressure and positively correlated with work engagement, respectively. Organizational change was not significantly related to employee innovation performance, and organizational identity was negatively related to work pressure (r = -0.363, p<0.001) and significantly positively related to organizational change (r = 0.084, p<0.05), work engagement (r = 0.653, p<0.001), and employee innovation performance (r = 0.396, p<0.001). The results of the correlation

**Table 1. Means, standard deviations, and correlations among study variables.**

| Variables | 1 | 2 | 3 | 4 | 5 | 6 | 7 | 8 | 9 | 10 | 11 |
|---|---|---|---|---|---|---|---|---|---|---|---|
| 1 gender | 1 | | | | | | | | | | |
| 2 age | -0.198*** | 1 | | | | | | | | | |
| 3 education | 0.023 | -0.083* | 1 | | | | | | | | |
| 4 job position | -0.011 | 0.308*** | 0.233*** | 1 | | | | | | | |
| 5 tenure | -0.229*** | 0.636*** | -0.290*** | 0.236*** | 1 | | | | | | |
| 6 company size | -0.327*** | 0.319*** | -0.112** | -0.089* | 0.557*** | 1 | | | | | |
| 7 OC | -0.092* | 0.328*** | -0.254*** | 0.374*** | 0.479*** | 0.301*** | *0.797* | | | | |
| 8 WE | 0.191*** | 0.099* | -0.028 | 0.152*** | 0.109** | 0.010 | 0.224*** | *0.927* | | | |
| 9 WP | 0.076 | 0.087* | -0.085* | 0.324*** | 0.049 | -0.216*** | 0.327*** | -0.200*** | *0.869* | | |
| 10 EIP | 0.091* | -0.099* | 0.070 | 0.076 | -0.110** | -0.114** | -0.003 | 0.491*** | -0.360*** | *0.921* | |
| 11 OI | 0.215*** | -0.006 | -0.017 | 0.080 | 0.001 | -0.030 | 0.084* | 0.653*** | -0.363*** | 0.396*** | *0.843* |

*p<0.05;

** p<0.01

*** p<0.001.

n = 289. The lower left matrix is the correlation coefficient, and the square root of the mean variance extracted is in the diagonal brackets. OC, organizational change; WE, work engagement; WP, work pressure; EIP, employee innovation performance; OI, organizational identity.

analysis indicated that the hypothesized relationships among organizational change, work pressure, work engagement, employee innovation performance, and organizational identity were initially verified. In addition, the correlation coefficients between the core variables were all less than 0.7, reflecting the absence of serious multicollinearity among the variables.

## The measurement model

We first examined the convergent and divergent validity of our measures. Specifically, a series of confirmatory factor analyses (CFA) and model comparisons were conducted. CFA analyses found that the five-factor (i.e., organizational change, work engagement, work pressure, employee innovation performance, and organization identity) measurement model fit the data well: $\chi^2$/df = 2.373, p<0.01, RMSEA = 0.069, SRMR = 0.058, TLI = 0.919, CFI = 0.925, (Table 2). All factor loading for items were significant (ps<0.01). Results of model

**Table 2. Confirmatory factor analysis results for the measures of the all variables.**

| Model and structure | $\chi^2$/df | RMSEA | SRMR | TLI | CFI |
|---|---|---|---|---|---|
| One-factor model | 8.678 | 0.163 | 0.139 | 0.545 | 0.571 |
| Two-factor model | 6.657 | 0.140 | 0.127 | 0.665 | 0.685 |
| Three-factor model | 4.047 | 0.103 | 0.107 | 0.819 | 0.831 |
| Four-factor model | 3.306 | 0.089 | 0.091 | 0.863 | 0.872 |
| Five-factor model: proposed | 2.373 | 0.069 | 0.058 | 0.919 | 0.925 |
| Six-factor model | 2.372 | 0.069 | 0.600 | 0.919 | 0.925 |

*One-factor Model* all variables combined

*Two-factor* work engagement, work pressure, employee innovation performance, organizational identity combined

*Three-factor Model* work pressure, employee innovation performance, organizational identity combined

*Four-factor Model* employee innovation performance, organizational identity combined

*Five-factor Model* hypothesized model

*Six-factor Model* five theoretical constructs with the latent common methods variance factor.

comparisons further demonstrated that the hypothesized five-factor measurement model had a significant better fit to the data than any of the alternative four-factor models (i.e., combining any two of the five factors). Such findings provided evidence of construct distinctiveness.

## Common method variance

Although we collect three-phase data, all five variables are from the same source. We use multiple methods, namely Harman's single-factor test and controlling for the ULMC (Unmeasured Latent Method Construct) methods factor to test for the presence of common method variance [52]. As shown in Table 2, a one-factor model does not fit well ($\chi^2$/df = 8.678, $p$>0.05; CFI = 0.571, TLI = 0.545, SRMR = 0.139 and RMSEA = 0.163), whereas the five-factor model fits satisfactorily ($\chi^2$/df = 2.373, $p$<0.01, TLI = 0.919, CFI = 0.925, SRMR = 0.058 and RMSEA = 0.069). The $\chi^2$ comparison shows that the one-factor model is significantly worse than the five-factor model.

Referring to the treatment of Podsakoff et al. [52], we also control for the effects of the ULMC methods factor. We construct one latent variable, *CMV (common method variance)*, by loading all observed indicators of the five theoretical variables. Hence, we develop a six-factor model that includes five theoretical variables and *CMV*. The results reveal that the six-factor model ($\chi^2$/df = 2.372, $p$<0.01, TLI = 0.919, CFI = 0.925, SRMR = 0.060 and RMSEA = 0.069) does not substantially improve the goodness of fit of the six-factor model (Table 2). Based on the above judgment, the effect of common method variance was not significant in this study.

## Hypothesis testing

This study uses SPSS 22.0 software to conduct a Bootstrap test on the moderating effect of organizational identification between the independent variable and the mediator, as well as the moderated mediation effect.

**Main effect test.**   Regression analysis was conducted on work pressure, work engagement and employee innovation performance to initially test the first half path and second half path effects of the mediating effects of work pressure and work engagement, and then Bootstrap test was applied to the dual mediating effects of juxtaposition using the Process procedure to further test the overall mediating effects and to improve robustness.

The results of the regression analysis between organizational change, work pressure, and employee innovation performance are shown in Table 3. By comparing model 1 and model 2, organizational change has a significant positive effect on work pressure (b = 0.322, p< 0.001) based on the control variables, thus hypothesis H1a is supported. Model 3 represents the impact of control variables on the dependent variable without the inclusion of independent variables. Model 4 represents the main effects, and Model 5 represents the impact on the dependent variable after the inclusion of mediating variables. Comparing model 4 and model 5, the effect of organizational change on employee innovation performance changed from insignificant to significant after adding mediating variables, and work pressure showed a significant negative predictive effect on employee innovation performance (b = -0.535, p< 0.001), whereby hypothesis H1b was supported.

The results of the regression analysis between organizational change, work engagement, and employee innovation performance are shown in Table 4. By comparing Model 1 and Model 2 in Table 4, on the basis of the control variables, organizational change has a significant positive effect on work engagement (b = 0.216, p< 0.001), i.e., organizational change positively promotes work engagement, thus hypothesis H2a is verified. Model 3 represents the impact of control variables on the dependent variable without the inclusion of independent variables. Model 4 represents the main effects, and Model 5 represents the impact on the dependent

**Table 3. Regression results for organizational change, work pressure on innovation performance.**

| Variable | WP | | EIP | | |
|---|---|---|---|---|---|
| | Model 1 | Model 2 | Model 3 | Model 4 | Model 5 |
| gender | 0.076 | 0.044 | 0.151 | 0.147 | 0.170* |
| age | 0.079 | 0.063 | -0.074 | -0.076 | -0.042 |
| education | -0.161* | -0.078 | 0.069 | 0.078 | 0.036 |
| job position | 0.559*** | 0.336*** | 0.174* | 0.147 | 0.327*** |
| tenure | -0.005 | -0.043 | -0.065 | -0.069 | -0.092 |
| company size | -0.184** | -0.249*** | -0.052 | -0.059 | -0.193*** |
| OC | | 0.322*** | | 0.038 | 0.210*** |
| WP | | | | | -0.535*** |
| $R^2$ | 0.180 | 0.244 | 0.034 | 0.035 | 0.252 |
| F | 15.637*** | 20.362*** | 2.531* | 2.256* | 19.073*** |

*$p < 0.05$
**$p < 0.01$
***$p < 0.001$.
n = 289.

variable after the inclusion of mediating variables. On the basis of model 4, with the addition of mediating variables, the results showed that the regression coefficient of organizational change on employee innovation performance increased but was not significant, and work engagement had a significant positive predictive effect on employee innovation performance (b = 0.545, p< 0.001), thus hypothesis H2b was verified.

In Tables 3 and 4, it is known that the $R^2$ is below 0.5, respectively. Although the thresholds for acceptable $R^2$ in the social sciences and organizational behavior disciplines are quite lenient, it is necessary to explain in the article how the explanatory power and its consequences manifest in the results of the proposed model. Firstly, a low $R^2$ value may reflect that there are variables not considered in the model that could significantly impact the dependent variable.

**Table 4. Regression results for organizational change, work engagement on innovation performance.**

| Variables | WE | | EIP | | |
|---|---|---|---|---|---|
| | Model 1 | Model 2 | Model 3 | Model 4 | Model 5 |
| gender | 0.424*** | 0.402*** | 0.151 | 0.147 | -0.072 |
| age | -0.009 | -0.020 | -0.074 | -0.076 | -0.065 |
| education | -0.145 | -0.090 | 0.069 | 0.078 | 0.127 |
| job position | 0.268* | 0.118 | 0.174* | 0.147 | 0.083 |
| tenure | 0.108 | 0.083 | -0.065 | -0.069 | -0.115* |
| company size | 0.062 | 0.018 | -0.052 | -0.059 | -0.069 |
| OC | | 0.216*** | | 0.038 | -0.080 |
| WE | | | | | 0.545*** |
| $R^2$ | 0.090 | 0.119 | 0.034 | 0.035 | 0.297 |
| F | 7.037*** | 8.514*** | 2.531* | 2.306* | 23.936*** |

*$p < 0.05$
**$p < 0.01$
***$p < 0.001$.
n = 289.

Secondly, this may also imply that the model has limited predictive power. Nonetheless, the model may still provide insights into certain key relationships. Furthermore, a low $R^2$ value may suggest that we need to further explore the data or consider using different models or methods to enhance explanatory power.

**Mediating effect test.** The control variables, independent variable organizational change, mediating variable work pressure, mediating variable work engagement, and dependent variable employee innovation performance were put into the Process program at the same time, and the mediation test model 4 was selected, the Bootstrap sampling number was set to 5000 (The number of bootstrap samples is set to 5000 because it provides sufficient precision and convergence while adhering to the empirical standards in social science research. This figure ensures the reliability of statistical estimates without unnecessary waste of computational resources), and the confidence level of the confidence interval was set to 95%, resulting in the path coefficient results shown in Table 5. According to the results in the table, the path coefficient of the effect of organizational change on work pressure is positive, while the path coefficient of the effect of work pressure on employee innovation performance is negative, and this result is significant at the 95% confidence interval level (all confidence intervals do not include 0), further verifying H1a and H1b. The path coefficient of the effect of organizational change on work engagement is positive, and the path coefficient of the effect of work engagement on employee innovation performance is also positive, and the result also show the indirect effects of work pressure and work engagement, where the effect of organizational change on employee innovation performance through work pressure is significantly negative and the effect of work engagement on employee innovation performance is significantly positive, both at the 95% confidence interval level. The results are significant at the 95% confidence interval level, indicating that there is a mediating effect of work pressure and work engagement between organizational change and employee innovation performance, i.e., hypotheses H1c and H2c are supported. So far, hypotheses H1a, H1b, H1c and hypotheses H2a, H2b, H2c were all supported.

**Moderating effect test.** Before conducting the moderating model runs, both the independent and moderating variables were centered to reduce the multidisciplinary of the variables (Table 6). In the work pressure regression model, model 2 showed significant interaction term regression coefficients after adding the independent variables, moderating variables, and the product of the two interaction terms based on model 1, indicating the moderating role of organizational identity in the process of organizational change affecting work pressure. Similarly, in the work engagement model, the regression coefficient of the interaction term between

**Table 5. Results of path coefficient.**

| Path | Coefficient | S.D. | p | 95% confidence intervals |
|---|---|---|---|---|
| OC→WP | 0.322*** | 0.047 | 0.000 | [0.2302, 0.4128] |
| WP→EIP | -0.369*** | 0.041 | 0.000 | [-0.4490, -0.2891] |
| OC→WE | 0.216*** | 0.050 | 0.000 | [0.1178, 0.3149] |
| WE→EIP | 0.420*** | 0.038 | 0.000 | [0.3455, 0.4936] |
| OC→WP→EIP | -0.119 | 0.029 | | [-0.1783, -0.0654] |
| OC→WE→EIP | 0.091 | 0.027 | | [0.0431, 0.1480] |

*$p<0.05$

**$p<0.01$

***$p<0.001$.

n = 289.

Table 6. Test of the moderating effect of organizational identification.

| Variables | WP | | WE | |
|---|---|---|---|---|
| | Model 1 | Model 2 | Model 3 | Model 4 |
| gender | 0.076 | -0.030 | 0.424*** | 0.472*** |
| age | 0.079 | 0.035 | -0.009 | 0.005 |
| education | -0.161* | -0.062 | -0.145 | -0.102 |
| job position | 0.559*** | 0.253* | 0.268* | 0.192* |
| tenure | -0.005 | -0.051 | 0.108 | 0.091 |
| company size | -0.184** | -0.224*** | 0.062 | -0.008 |
| OC | | 0.261*** | | 0.274*** |
| OI | | -0.117*** | | 0.096* |
| OC ×OI | | -0.194*** | | 0.186*** |
| $R^2$ | 0.180 | 0.303 | 0.090 | 0.171 |
| F | 15.637*** | 22.384*** | 7.037*** | 10.584*** |

*$p<0.05$
**$p<0.01$
***$p<0.001$.
n = 289.

organizational change and organizational identity is significant in model 4, indicating that organizational identity moderates the effect of organizational change on work engagement.

Based on the model of the dual mediating path effect of organizational change on employee innovation performance, adding the moderating variable organizational identity, the overall mediating-regulating effect path can be obtained as shown in Figs 2 and 3. Under the moderating effect, organizational change significantly and positively predicts work pressure and work engagement, but organizational identity negatively moderates the effect of organizational change on work pressure and positively moderates the effect of organizational change on work engagement, respectively.

To further explain the moderating effect of organizational identity, simple slope estimation was conducted in this study (see Figs 2 and 3). As shown in Fig 2, the negative effect of organizational change on work pressure is stronger at low organizational identity compared to high organizational identity level, so organizational identity can somewhat weaken the negative predictive effect of organizational change on work pressure. Fig 3 shows that the positive effect of organizational change on work engagement is stronger at high organizational identity levels. Thus, hypotheses H3a and H3b are supported.

## Overall moderated mediation model

The mediating effect with moderation was tested using the bootstrapping method (Table 7). Under the mediated path of work pressure, low organizational identity enhances the negative effect of organizational change on work pressure, which in turn reduces employee innovation performance. High organizational identification weakened the above influence process, but the effect was not significant. In the mediated path of work engagement, low organizational identity weakened the effect of organizational change on work engagement, but the effect did not reach significance at the 95% confidence level. High organizational identity significantly enhances the positive effect of organizational change on work engagement, which in turn improves employee innovation performance. The mediating effects with moderation in the

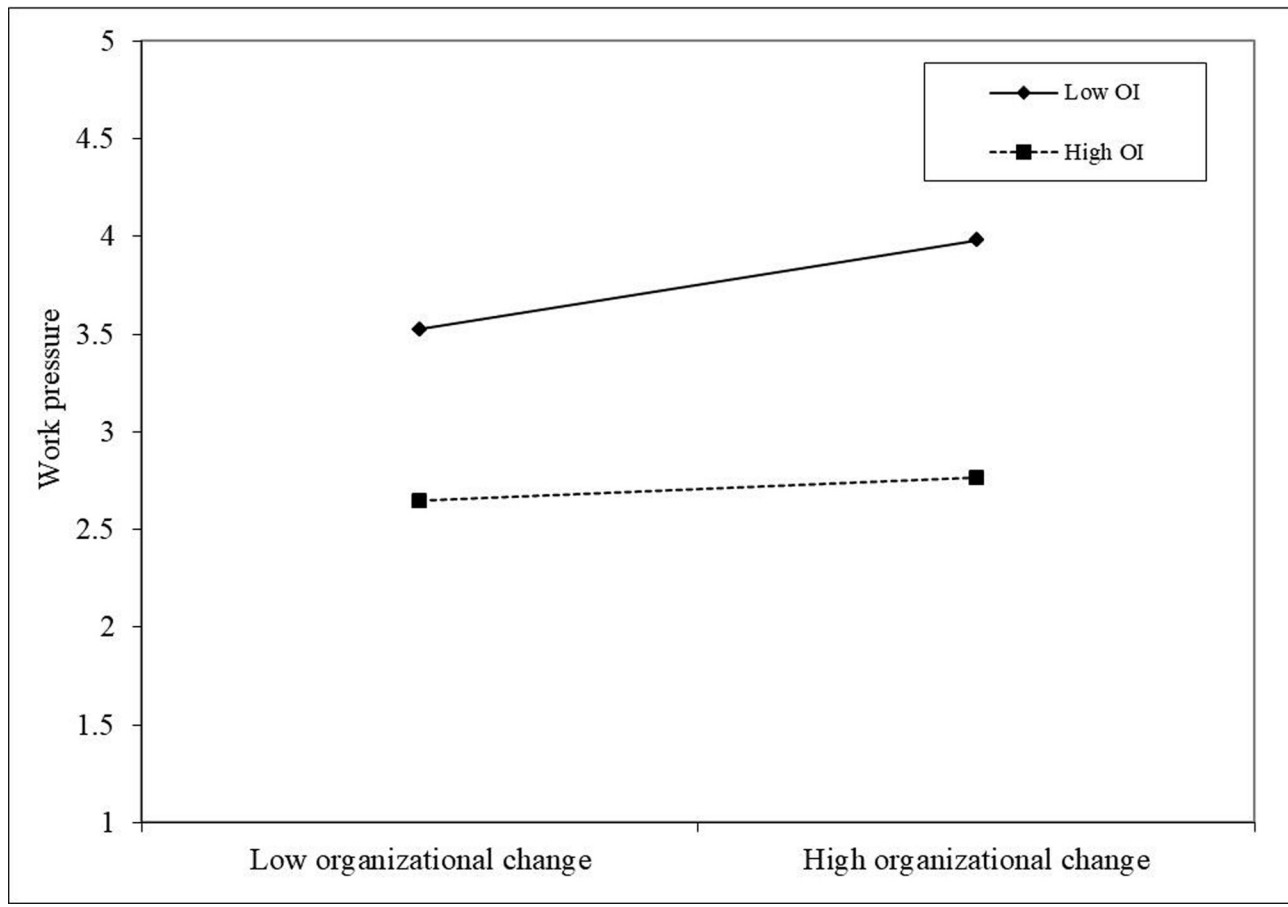

**Fig 2. Moderating effect of organizational identity on the relationship between organizational change and work pressure.**

dual mediation model were significantly positive (all 95% confidence intervals did not include 0), so hypotheses H4a and H4b were supported.

## Discussion

Based on the JD-R model, this study investigates the "double-edged sword" effect of organizational change on employees' innovation performance, and considers the boundary role of employee-organization relationship in the above influence process.

First, organizational change negatively affects employee innovation performance through the mediating role of work pressure. Empirical studies show that organizational change positively affects work pressure, while work pressure negatively affects employee innovation performance. According to JD-R model theory, any job characteristic can be divided into job demands and job resources, in which job demands can consume employees physically and mentally and thus lead to health depletion process and eventually lead to negative job outcomes [53]. The increased work demands due to change, such as workload, role conflict, and organizational politics, tend to consume more energy and time for employees to complete their old or new work tasks, thus increasing their work pressure [54]. In addition, a series of new policies and events in the change may require employees to recogncance and re-adapt, and the uncertainty brought by the change makes them psychologically worried and apprehensive, thus creating a sense of insecurity and increasing work pressure. Employees under high

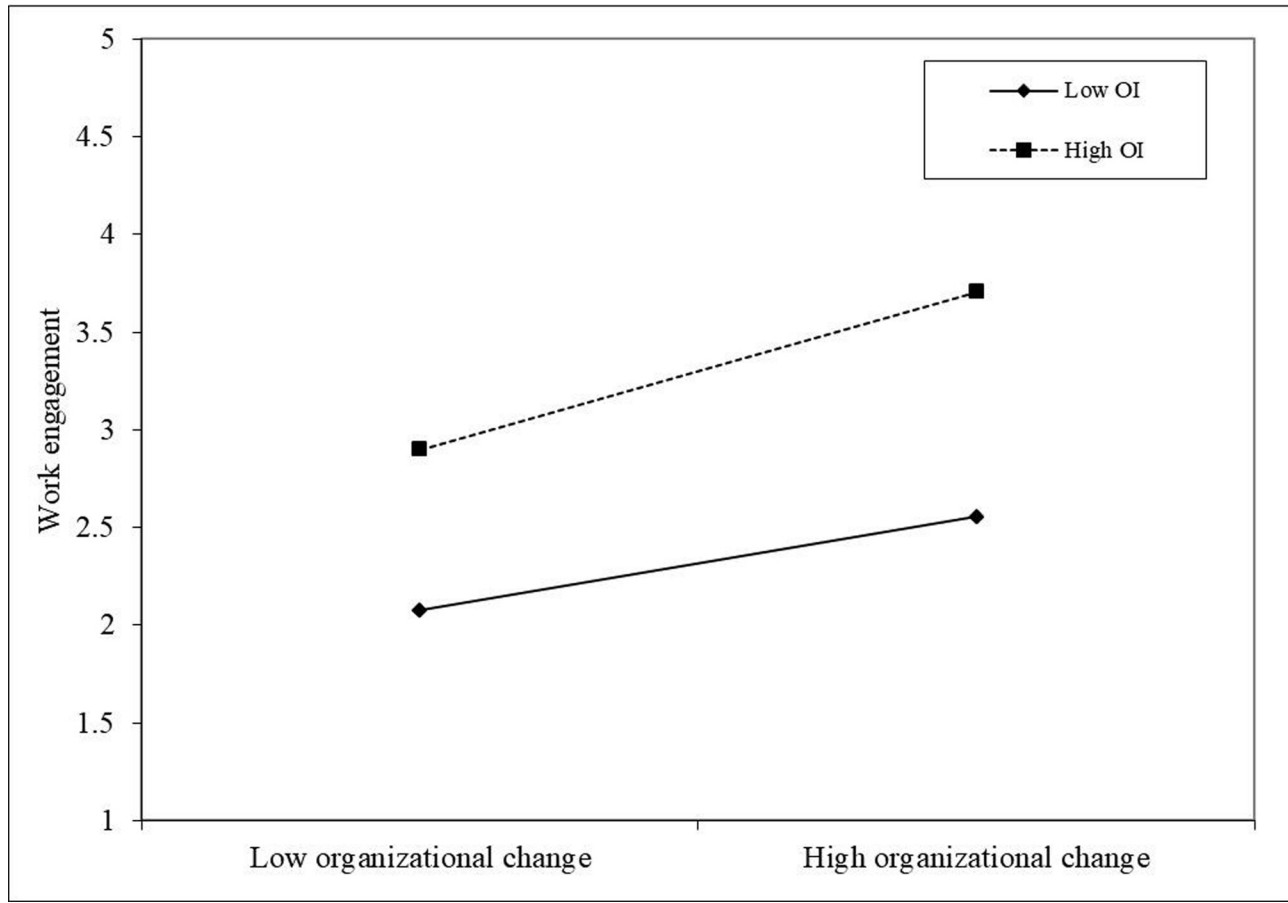

**Fig 3. Moderating effect of organizational identity on the relationship between organizational change and work engagement.**

work pressure tend to focus more on dealing with negative emotions and stress to seek "resource protection" during the change phase, so it is difficult to devote time to improving work effectiveness and personal innovation, which ultimately leads to low innovation performance.

Second, organizational change positively affects employee innovation performance through the mediating role of work engagement. The empirical results show that organizational change positively affects work engagement, and work engagement positively affects employee innovation performance; therefore, work engagement positively mediates the relationship between organizational change and employee innovation performance. In the process of organizational

**Table 7. Results of moderated mediation effect test.**

| Mediator | Moderator | Coefficient | 95% confidence intervals |
|---|---|---|---|
| WP | Low OI | -0.156 | [-0.2252, -0.0923] |
| | High OI | -0.037 | [-0.0954, 0.0145] |
| | moderated mediation effect | 0.072 | [0.0329, 0.1100] |
| WE | Low OI | 0.050 | [-0.0025, 0.1152] |
| | High OI | 0.180 | [0.1131, 0.2479] |
| | moderated mediation effect | 0.078 | [0.0283, 0.1223] |

change, procedural fairness, support from superiors, change participation and feedback are all work resources that are added to employees' work. These work resources create a favorable change climate, help employees better understand and support change, and allow them to perceive higher work value and stronger work meaning, so that they are more engaged in their work and stimulate their personal creativity to perform better in innovation at work performance.

Finally, organizational identity negatively moderates the negative effect of organizational change on work pressure and positively moderates the positive effect of organizational change on work engagement, while moderating the mediating role of work pressure and work engagement between organizational change and employee innovation performance. The results of the study show that in the mediated-regulated model, organizational identity has a significant moderating role in the process of organizational change affecting work pressure and work engagement, and low organizational identity amplifies the negative effect of organizational change on work pressure, while high organizational identity enhances the positive effect of organizational change on work engagement. The moderating effect of organizational identity on the two mediating variables was also significant in the test of the mediating model with moderation. According to social identity theory, individuals with identity are usually consistent with the society or group to which they belong in many aspects of behavior and perceptions. Organizational identity is an extension of social identity in the organizational environment, and employees with a high level of organizational identity tend to rely on the organization, trust the organization, align their personal goals and interests with those of the organization, and are willing to do their best for the organization [55]. Therefore, in the face of the "test" of organizational change, members with high organizational identity can also support and cooperate with organizational decisions, actively participate in the practice of change, become more involved in their work, and actively mobilize their personal initiative, stimulate their creativity, and better promote the development of change. Employees with low organizational identity, on the other hand, do not have strong emotional ties with the organization, and are more inclined to respond negatively when they encounter negative effects such as work requirements brought about by change, lack trust in the organization, and are more likely to focus on the potential risks of change, thus generating negative emotions and feeling greater pressure. Out of the protection of their own resources, employees under high pressure are more inclined to try to maintain the timely completion of current work tasks, rather than spending more energy and time to come up with new work ideas, create new work methods, etc., thus showing lower innovative performance.

Above all, under the influence of the VUCA environment, our research findings reveal that the impact of organizational change on employee innovation performance may fluctuate in complex ways. For instance, in conditions of high volatility, employees may struggle to adapt to frequent changes, which could diminish their innovation performance. Uncertainty might hinder employees' clear understanding of organizational goals and direction, thereby reducing their motivation to drive innovation. Complexity increases the unpredictability of the work environment, potentially distracting employees and limiting their capacity and time to engage in exploratory work. Ambiguity may lead to confusion over expectations and responsibilities, thereby decreasing their efforts on innovation projects. The possible reasons behind these changes include differences in employees' adaptability to unstable environments, insufficient internal communication, ineffective change management strategies, and inadequate support from the organizational culture for change and innovation. Therefore, organizations need to develop and implement effective strategies to cope with the VUCA environment, in order to stimulate employees' innovative potential and enhance their performance.

## Theoretical implications

First, based on the JD-R model theory, the "double-edged sword" effect of organizational change on employees' innovation performance is confirmed at the individual subjective level. Since the emergence of organizational change, experts and scholars have been focusing on the impact of this context on employees' work. Given the influence and importance of employees' attitudes on the outcome of change, more studies have focused on employees' attitudes and behaviors toward change, but less on the impact of organizational change on employees' work outcomes in normal work situations. Innovation has always been the main theme of corporate development, and employee innovation performance to some extent reflects organizational vitality and innovation base. Existing studies have investigated the antecedent variables of employee innovation performance in terms of individual factors, job factors, and factors in regular organizational contexts, while less attention has been paid to the current increasingly frequent organizational change contexts. The only relevant literature, too, has derived the effects of organizational change on employee innovation in terms of positive or negative aspects, respectively, while ignoring attempts to explore both positive and negative opposing effects based on an integrative perspective. From the perspective of the JD-R model, this study argues for the dual mediating role of work pressure and work engagement between organizational change and employee innovation performance, taking the changes in job requirements and job resources due to organizational change as triggers, respectively, and provides empirical evidence for the simultaneous positive and negative effects of organizational change on employee innovation performance.

Second, embedding the JD-R model in an organizational change context expands the scope of application of the JD-R model and enriches the research on the results of the JD-R model. However, the number of studies that have embedded the model in a particular organizational context is very small. In fact, the JD-R model provides a very good perspective and theoretical support in exploring the topic of organizational change affecting employees' innovative performance. These changes have negative and positive effects through the dual path theory of the JD-R model, which explains the "double-edged sword" effect of organizational change on employees' innovation performance. In addition, job happiness and job performance are the variables that have been widely focused on in the JD-R model's work outcome studies, while innovation as an outcome variable is relatively uncommon in JD-R-related studies.

Finally, based on social identity theory, the boundary role of organizational identity in the dual mediating mechanism of organizational change on employee innovation performance was explored and confirmed. It was found that organizational change can produce very different results on employee innovation performance through different mediating mechanisms, and this process is also moderated by organizational identity. The higher the degree of employees' organizational identification, the more they can show positive support for organizational decisions and actions, and the more they are willing to put in more efforts and attempt to help achieve organizational goals. Therefore, when facing organizational change events with uncertainty, organizational identification may directly influence employees' change responses and behavioral choices. To a certain extent, the above discussion reveals the important role of employee-organization relationship in the process of organizational change affecting employees' work outcomes that cannot be ignored, and also expands the application of social identity theory in specific organizational development contexts and enriches the content of organizational identity theory.

## Practical implications

First, pay attention to the work requirements and resource changes brought about by organizational change. "Organizational change is a series of innovation from top to bottom, and the

successful implementation of this "big move" is actually inseparable from the active participation and support of every member of the organization. JD-R model theory provides a novel perspective to help managers effectively guide employees to actively participate in the change process. The changes in employees' work characteristics should be categorized and managed to balance the incremental changes in employees' work requirements and work resources. On the one hand, leaders of each department should strengthen effective communication with employees, always synchronize the systems and policies in the change, increase the training related to organizational change, clarify the new organizational structure and the specific job responsibilities and goals of employees in the change, reduce employees' perception of job requirements, and relieve employees' work pressure. On the other hand, management should strengthen the work resources in the change, establish open and transparent communication channels between top and bottom, maintain procedural justice in the change, clearly communicate to employees that they support their participation in organizational change and their personal efforts for the change, and encourage and give positive feedback to employees' positive change performance in a timely manner to enhance employees' sense of work meaning and increase their work commitment.

Second, pay attention to the dual-path impact of organizational change on employees' innovation performance. Companies have been sparing no effort to stimulate employees' innovation potential, such as improving leadership style and creating a good innovation atmosphere, to enhance employee innovation and thus improve the overall innovation performance of the organization. In fact, in addition to the targeted improvement of innovation management methods and tools, special organizational situations may also bring "unexpected benefits". Although organizational change is primarily designed to adapt to changes in the internal and external environment, improve organizational efficiency, and better achieve organizational goals, not to improve employee innovation, organizational change itself is a collection of new things and events that provide a "breeding ground" for employee innovation. This study has confirmed that organizational change may hinder employee innovation through work pressure or promote employee innovation through work engagement. As a company manager, you can focus your efforts on improving the positive impact of organizational change on employee innovation by paying attention to and satisfying employees' reasonable needs, striving to improve employees' work engagement, actively implementing innovation incentive policies, organizing relevant innovation practice activities, and encouraging employees to They can focus on improving the positive impact of organizational change on employees' innovation, meeting employees' reasonable needs, improving employees' commitment to work, actively implementing innovation incentive policies, organizing relevant innovation practice activities, encouraging employees to give full play to their initiative and creativity in the change situation, cultivating employees' innovation consciousness and stimulating their innovation potential to improve their innovation performance.

Finally, focus on developing and improving employees' organizational identity. Research findings show that the double-edged sword effect of organizational change on employees' innovation performance is power-variable, and low organizational identity will increase the negative impact of organizational change and offset its positive impact to a certain extent. In Sun Tzu's "The Art of War", it is mentioned that "the one who wants the same thing from the top and the bottom wins", and employees with a high sense of organizational identity are often able to "unite with the organization" and form "the same desire from the top and the bottom" to work together to achieve the goals. Therefore, it is very important to cultivate employees' sense of organizational identity. In recruiting employees, HR should pay attention to the match between employees' values and the organization's cultural values, and "fasten the first button of organizational identity" when new employees join the company. Further, employee

organizational identity is not static and cannot be achieved overnight, but can be cultivated and improved by the company through later efforts. Internally, enterprises should create a good working environment with transparent policies and fair systems in their daily management to ensure fair benefits for employees and win their trust in the organization, while paying attention to the quality improvement of managers, strengthening harmonious communication between management and grassroots employees, and encouraging leaders to increase humanistic care for employees at work and their family life behind them. Externally, enterprises should take more social responsibility in their business philosophy, enhance the organization's reputation, create a positive brand image, promote employees and the organization in the rational and emotional "two-way run", strengthen employees' sense of organizational identity, so that employees from the heart to link their personal interests with the interests of the organization, and thus strive to create performance for the organization The brand image of the organization can be strengthened.

## Limitations

This study has followed the scientific procedures in model formulation and research design, but there are still some shortcomings due to our professional ability, resource conditions and objective limitations: (1) Due to geographical limitations, this study only selected some enterprises in Shandong in the questionnaire survey, and there may be differences in the empirical results of enterprises in different regions in this topic. Future research can broaden the questionnaire distribution channels, expand the sample area, and further analyze with a more random, diverse, and universal sample to improve the generalization of the study. (2) The questionnaire adopted in this study was completed by employees' self-assessment. Although the three-stage questionnaire design reduces homogeneous error to a certain extent, the influence of personal subjective factors on the study cannot be completely excluded. Future research can try questionnaire other assessment, up and down paired survey, etc. to further reduce homogenization error through more objective data. (3) In terms of the timing of the questionnaire, the surveyed companies were undergoing organizational change during the research process, so the data reported by employees during the three stages of the questionnaire before and after about 1.5 months came from after the organizational change and did not include the data before the organizational change, and the relationship between the relevant variables may not be convincing. Future research could follow up with companies to cover the research process before and after organizational change, and further explore the differences in the impact of organizational change on employees by comparing data from different stages.

## Supporting information

**S1 File. Data.**
(XLS)

## Author Contributions

**Conceptualization:** Lei Ning.

**Data curation:** Lei Ning, Daokui Jiang.

**Formal analysis:** Teng Liu, Lei Ning.

**Funding acquisition:** Hao Wang, Zhenzhu Li, Lei Ning.

**Investigation:** Teng Liu, Hao Wang, Yaru Liu, Zhenzhu Li, Yiting Zhang, Honghong Zhu, Lei Ning, Daokui Jiang.

**Resources:** Honghong Zhu, Lei Ning.

**Writing – original draft:** Teng Liu, Yiting Zhang, Lei Ning.

**Writing – review & editing:** Yaru Liu, Honghong Zhu, Lei Ning.

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
