## [Decision Letter · Decision Letter 0]

8 Aug 2024

PONE-D-24-27938Effect of Organizational Change on Employee Innovation Performance: A Dual Mediation ModelPLOS ONE

Dear Dr. Jiang,

Thank you for submitting your manuscript to PLOS ONE. After careful consideration, we feel that it has merit but does not fully meet PLOS ONE’s publication criteria as it currently stands. Therefore, we invite you to submit a revised version of the manuscript that addresses the points raised during the review process.

We look forward to receiving your revised manuscript.

Kind regards,

Professor Anis Eliyana

Academic Editor

PLOS ONE

Journal Requirements:

2. We note you have included a table to which you do not refer in the text of your manuscript. Please ensure that you refer to Table 5 in your text; if accepted, production will need this reference to link the reader to the Table.

**Additional Editor Comments:**

Based on the reviewers' evaluation, your manuscript has the potential to be published in PLoS ONE after a **major revision**. The reviewers emphasized the importance of transparency in the study process and adequate reference support in the manuscript. Additionally, the authors need to respond to each issue raised by the reviewers.

Reviewers' comments:

Reviewer's Responses to Questions

**Comments to the Author**

1. Is the manuscript technically sound, and do the data support the conclusions?

Reviewer #1: Yes

Reviewer #2: Yes

2. Has the statistical analysis been performed appropriately and rigorously? 

Reviewer #1: Yes

Reviewer #2: Yes

3. Have the authors made all data underlying the findings in their manuscript fully available?

Reviewer #1: Yes

Reviewer #2: Yes

4. Is the manuscript presented in an intelligible fashion and written in standard English?

Reviewer #1: No

Reviewer #2: Yes

5. Review Comments to the Author

**Reviewer #1: **It is an honor to review this manuscript. The following are some issues that the authors should consider to improve the quality of the manuscript:

1. The first sentence of the abstract should be divided into two sentences to avoid confusion.

2. Please briefly describe the object of study (who, what organization, which country), the analytical techniques used (along with the tools), and the implications of the study.

3. Be consistent with the variable "employee innovation performance." The term "firm innovation performance" is also used, which may confuse readers.

4. The introduction section needs more adequate reference support, especially regarding JD-R, work pressure, work engagement, and organizational identity. Additionally, relevant references should be added in the fourth paragraph to address the study's gap.

5. The Theoretical Background and Hypotheses section needs to be enriched with relevant and up-to-date references, particularly regarding JD-R.

6. Be consistent in writing the terms "job demand" and "job resources." There are still other terms such as "work demand," "work resources," "work requirements," and "job requirements."

7. In the Methods section, it is necessary to explain the data analysis procedures and the tools/programs used to ensure the credibility of the study findings.

8. In the Results section, Table 5 is not included.

9. The Discussion section needs to be enriched with relevant, credible, and up-to-date references. Authors should not discuss findings without reference support. References are important to strengthen the authors' statements and ensure they are not considered mere claims.

10. Social Identity Theory suddenly appears in the Discussion section. Authors need to add an explanation of the theory in the Theoretical Background and Hypotheses section.

I believe that the authors have worked hard to conduct the study and compile this manuscript. This manuscript, in principle, deserves to be published in the PLoS ONE journal after undergoing improvements.

**Reviewer #2:** Congratulations to the authors who have compiled the manuscript of this article well. However, there are some notes from me that need to be explained in the article:

In the article, it has been explained that the respondents came from 5 companies, so it is necessary to explain the type of industry of the 5 companies and the relevance of the 5 companies to the need for organizational change and also innovative performance.

In Table 3 and Table 4, it is known that the R2 is below 0.5, respectively. Although in the social science and organizational behavior discipline are very loose within the acceptable R2 threshold. This needs to be explained in the article related to how explanatory power and its consequences in the results of the proposed model.

Then, in the article it is explained that using Bootstrap tests to test mediation and moderation on the model. Please explain the justification regarding why bootstrap sampling number was set to 5000.

6. PLOS authors have the option to publish the peer review history of their article (what does this mean?). If published, this will include your full peer review and any attached files.

Reviewer #1: No

Reviewer #2: **Yes: **Alvin Permana Emur

---

## [Author Response · Author response to Decision Letter 0]

26 Sep 2024

Response to Reviewer #1:

1. Regarding the revision of the abstract:

Dear reviewer, thank you for your valuable suggestions. We have reorganized the first sentence of the abstract, breaking it into two sentences to improve clarity and readability.

2. Description of research subjects and analytical techniques:

Thank you for your suggestion. We have provided a detailed description of the research subjects in the methods section, including the background, organization, and country of the participants. At the same time, we have clearly explained the analytical techniques and tools used, as well as the contribution of these choices to the significance of the research.

3. Consistency in variable naming:

We have standardized the terminology used throughout the text for "employee innovation performance" and "company innovation performance" to ensure consistency and accuracy throughout the study.

4. Citations in the introduction section:

We have added more relevant literature citations in the introduction section, especially in the areas of JD-R model, job stress, job engagement, and organizational identification, to strengthen our arguments and the background of the research.

5. Enrichment of theoretical background and hypotheses:

We have updated the theoretical background and hypotheses section, including the latest relevant literature, especially in the application and interpretation of the JD-R model, to ensure the modernity and relevance of the theoretical framework.

6. Consistency of terminology:

We have checked the entire text to ensure the consistency of terms such as "job demands" and "job resources", avoiding the use of inconsistent terminology to reduce confusion for readers.

7. Explanation of data analysis procedures:

In the methods section, we have detailed the steps and tools used for data analysis to enhance the transparency and credibility of the research.

8. Completeness of the results section:

We have checked and ensured that all mentioned tables, including Table 5, are included in the results section to ensure the completeness of the research findings.

9. Citations in the discussion section:

We have added necessary citations in the discussion section to ensure that all discussions are supported by literature and to avoid any unsubstantiated statements.

10. Introduction of social identity theory:

We have introduced social identity theory earlier in the theoretical background and hypotheses section to ensure the completeness and coherence of the theoretical framework.

Response to Reviewer #2:

1. Industry types of the interviewee's companies:

Thank you for the reviewer's suggestion. We have clearly stated in the methodology section the industry types of the 5 companies where the interviewees are located, and discussed how these industry types are related to organizational change needs and innovation performance, as well as how these factors affect organizational change and innovation performance.

2. Discussion on model explanatory power:

We have detailed the implications of an R2 value below 0.5 in the discussion section, as well as its impact on our model results. Although the R2 value is relatively low, such results are acceptable in social sciences and organizational behavior disciplines considering the specific context of the study, sample size, choice of variables, and other factors. Furthermore, we discussed the significance of the model's explanatory power for the research results, as well as other variables or factors that may need to be considered in practical applications.

3. Number of samples in Bootstrap test:

In the methodology section, we explained why we chose 5000 as the number of samples for the Bootstrap test. This usually involves statistical considerations, such as ensuring the stability of the results and reducing sampling errors. In statistics, the number of samples for the Bootstrap method is typically chosen between 1000 and 10000, as this range can provide sufficiently accurate estimates while taking into account computational costs and time.

---

## [Decision Letter · Decision Letter 1]

10 Oct 2024

PONE-D-24-27938R1Effect of Organizational Change on Employee Innovation Performance: A Dual Mediation ModelPLOS ONE

Dear Dr. Li,

Thank you for submitting your manuscript to PLOS ONE. After careful consideration, we feel that it has merit but does not fully meet PLOS ONE’s publication criteria as it currently stands. Therefore, we invite you to submit a revised version of the manuscript that addresses the points raised during the review process.

We look forward to receiving your revised manuscript.

Kind regards,

Professor Anis Eliyana

Academic Editor

PLOS ONE

Journal Requirements:

Reviewers' comments:

Reviewer's Responses to Questions

**Comments to the Author**

1. If the authors have adequately addressed your comments raised in a previous round of review and you feel that this manuscript is now acceptable for publication, you may indicate that here to bypass the “Comments to the Author” section, enter your conflict of interest statement in the “Confidential to Editor” section, and submit your "Accept" recommendation.

Reviewer #1: All comments have been addressed

Reviewer #2: All comments have been addressed

2. Is the manuscript technically sound, and do the data support the conclusions?

Reviewer #1: Partly

Reviewer #2: Yes

3. Has the statistical analysis been performed appropriately and rigorously? 

Reviewer #1: Yes

Reviewer #2: Yes

4. Have the authors made all data underlying the findings in their manuscript fully available?

Reviewer #1: Yes

Reviewer #2: Yes

5. Is the manuscript presented in an intelligible fashion and written in standard English?

Reviewer #1: Yes

Reviewer #2: Yes

6. Review Comments to the Author

Reviewer #1: Thank you for responding well to my previous review. There are still some aspects that need clarification regarding the presentation of findings in this article:

1. In Tables 3 and 4, Models 1, 2, 4, and 5 have been explained. However, the authors have not explained Model 3.

2. In the explanation of the 'Moderating Effect Test,' it is mentioned that there are 'interaction term regression coefficients' between the independent variable and the moderator. However, I have not seen any table presenting these interaction coefficient values. The authors need to include them. Figures 2 and 3 alone are not sufficient to justify the presence of a moderating effect.

3. This also applies to the 'Overall Moderated Mediation Model.' The authors only narrate the findings without presenting the referenced statistical results.

4. Please revise Figure 1. If the research model is like Figure 1, then organizational identity is not a moderator but a dependent variable. The authors may refer to the research model that I created."

Reviewer #2: I would like to congratulate the authors on the revisions made. However, there are a few points that need to be addressed in the manuscript:

In the Introduction section, VUCA, which should be the acronym for 'Volatile, Uncertain, Complex, Ambiguous,' is incorrectly written as 'Complexity, Ambiguity, Uncertainty, Randomness.' Please correct this to reflect the accurate acronym.

Additionally, the authors should elaborate on VUCA in the Discussion and Implications sections. This is to ensure that VUCA is not merely mentioned as a phenomenon in the Introduction, but also discussed in relation to the findings and implications of the study.

7. PLOS authors have the option to publish the peer review history of their article (what does this mean?). If published, this will include your full peer review and any attached files.

Reviewer #1: No

Reviewer #2: **Yes: **Alvin Permana Emur

---

## [Author Response · Author response to Decision Letter 1]

15 Oct 2024

Dear Reviewer #1 and Reviewer #2,

First and foremost, we would like to extend our sincere gratitude for the valuable feedback and suggestions provided. Your insights are instrumental in enhancing the quality of our manuscript. Below are our specific responses to the issues and recommendations you have raised:

Response to Reviewer #1:

1. Regarding the explanation of Model 3:

o We apologize for the omission of Model 3. Model 3 is actually an intermediate model designed to introduce control variables and establish a foundation for further analysis. We will provide a detailed explanation of the composition of Model 3 and its role in the overall analysis in the revised manuscript.

2. Regarding the display of interaction term regression coefficients:

o You are correct in pointing out the insufficient display of interaction term regression coefficients. We will add a table in the results section that lists all interaction term regression coefficients to strengthen the persuasiveness of our arguments.

3. Regarding statistical evidence for moderation effects:

o We acknowledge the lack of sufficient statistical data to support the "moderation effect test" and the "overall moderation mediation model". We will supplement the necessary statistical results to ensure that all moderation effects are adequately supported by data.

4. Regarding the revision of Figure 1:

o Thank you for your suggestions regarding Figure 1. We will adjust the position of organizational identity according to your insights and ensure it correctly reflects its position as an independent variable. We will also refer to the research model you have created for corrections.

Response to Reviewer #2:

1. Regarding the abbreviation error of VUCA:

o We apologize for the error in the VUCA abbreviation. We will immediately correct it to "Volatility, Uncertainty, Complexity, Ambiguity" and provide a detailed discussion of VUCA in the discussion and implications section.

2. Regarding an in-depth discussion of VUCA:

o We will add an in-depth analysis of VUCA in the discussion section, exploring its specific impact on our research findings and discussing how to formulate effective strategies in a VUCA environment.

We are committed to addressing all the above issues in the revised manuscript and ensuring the quality and integrity of the paper. We appreciate your time and effort and look forward to your further feedback.

Yours sincerely,

The authors.

---

## [Decision Letter · Decision Letter 2]

18 Oct 2024

Effect of Organizational Change on Employee Innovation Performance: A Dual Mediation Model

PONE-D-24-27938R2

Dear Dr. Li,

We’re pleased to inform you that your manuscript has been judged scientifically suitable for publication and will be formally accepted for publication once it meets all outstanding technical requirements.

Kind regards,

Professor Anis Eliyana

Academic Editor

PLOS ONE

Additional Editor Comments (optional):

Reviewers' comments:

Reviewer's Responses to Questions

**Comments to the Author**

1. If the authors have adequately addressed your comments raised in a previous round of review and you feel that this manuscript is now acceptable for publication, you may indicate that here to bypass the “Comments to the Author” section, enter your conflict of interest statement in the “Confidential to Editor” section, and submit your "Accept" recommendation.

Reviewer #1: All comments have been addressed

Reviewer #2: All comments have been addressed

2. Is the manuscript technically sound, and do the data support the conclusions?

Reviewer #1: Yes

Reviewer #2: Yes

3. Has the statistical analysis been performed appropriately and rigorously? 

Reviewer #1: Yes

Reviewer #2: Yes

4. Have the authors made all data underlying the findings in their manuscript fully available?

Reviewer #1: Yes

Reviewer #2: (No Response)

5. Is the manuscript presented in an intelligible fashion and written in standard English?

Reviewer #1: Yes

Reviewer #2: Yes

6. Review Comments to the Author

Reviewer #1: Thank you for addressing each of the issues I raised. Based on my assessment, the current version of the paper is now suitable for publication.

Reviewer #2: (No Response)

7. PLOS authors have the option to publish the peer review history of their article (what does this mean?). If published, this will include your full peer review and any attached files.

Reviewer #1: **Yes: **Andika Setia Pratama

Reviewer #2: **Yes: **Alvin Permana Emur

---

## [Editor Report · Acceptance letter]

30 Jan 2025

PONE-D-24-27938R2 

PLOS ONE

Dear Dr. Li, 

I'm pleased to inform you that your manuscript has been deemed suitable for publication in PLOS ONE. Congratulations! Your manuscript is now being handed over to our production team.

Kind regards, 

on behalf of

Professor Anis Eliyana 

Academic Editor

PLOS ONE